# Non-Invasive Tools in Perioperative Stroke Risk Assessment for Asymptomatic Carotid Artery Stenosis with a Focus on the Circle of Willis

**DOI:** 10.3390/jcm13092487

**Published:** 2024-04-24

**Authors:** Balázs Lengyel, Rita Magyar-Stang, Hanga Pál, Róbert Debreczeni, Ágnes Dóra Sándor, Andrea Székely, Dániel Gyürki, Benjamin Csippa, Lilla István, Illés Kovács, Péter Sótonyi, Zsuzsanna Mihály

**Affiliations:** 1Department of Vascular and Endovascular Surgery, Heart and Vascular Center, Semmelweis University, 1122 Budapest, Hungary; lengyel.balazs92@gmail.com (B.L.); sotonyi@hotmail.com (P.S.J.); 2Department of Neurology, Semmelweis University, 1085 Budapest, Hungary; stang.rita@semmelweis.hu (R.M.-S.); pal.hanga@semmelweis.hu (H.P.); debreczeni.robert@med.semmelweis-univ.hu (R.D.); 3Szentágothai Doctoral School of Neurosciences, Semmelweis University, 1085 Budapest, Hungary; 4Department of Anesthesiology and Intensive Therapy, Semmelweis University, 1085 Budapest, Hungary; sandoragnesdora@gmail.com (Á.D.S.); szekely.andrea1@med.semmelweis-univ.hu (A.S.); 5Department of Hydrodynamic Systems, Faculty of Mechanical Engineering, Budapest University of Technology and Economics, 1085 Budapest, Hungary; dgyurki@hds.bme.hu (D.G.); bcsippa@hds.bme.hu (B.C.); 6Department of Ophthalmology, Semmelweis University, 1085 Budapest, Hungary; lilla.istvan@gmail.com (L.I.); kovacs.illes@med.semmelweis-univ.hu (I.K.); 7Department of Ophthalmology, Weill Cornell Medical College, New York, NY 10065, USA; 8Department of Clinical Ophthalmology, Faculty of Health Sciences, Semmelweis University, 1085 Budapest, Hungary

**Keywords:** asymptomatic carotid artery stenosis, Circle of Willis, cerebrovascular reserve capacity, non-invasive diagnostics tools, perioperative stroke risk

## Abstract

This review aims to explore advancements in perioperative ischemic stroke risk estimation for asymptomatic patients with significant carotid artery stenosis, focusing on Circle of Willis (CoW) morphology based on the CTA or MR diagnostic imaging in the current preoperative diagnostic algorithm. Functional transcranial Doppler (fTCD), near-infrared spectroscopy (NIRS), and optical coherence tomography angiography (OCTA) are discussed in the context of evaluating cerebrovascular reserve capacity and collateral vascular systems, particularly the CoW. These non-invasive diagnostic tools provide additional valuable insights into the cerebral perfusion status. They support biomedical modeling as the gold standard for the prediction of the potential impact of carotid artery stenosis on the hemodynamic changes of cerebral perfusion. Intraoperative risk assessment strategies, including selective shunting, are explored with a focus on CoW variations and their implications for perioperative ischemic stroke and cognitive function decline. By synthesizing these insights, this review underscores the potential of non-invasive diagnostic methods to support clinical decision making and improve asymptomatic patient outcomes by reducing the risk of perioperative ischemic neurological events and preventing further cognitive decline.

## 1. Introduction

In the European population of 715 million, strokes afflict approximately 1.4 million individuals annually, with 1.1 million succumbing to stroke-related mortality each year. A transient ischemic attack (TIA) is defined as a sudden onset of focal neurological dysfunction lasting <24 h, which is of a non-traumatic, vascular origin. Conversely, a stroke is defined as a sudden onset of focal (rather than global) onset of neurological dysfunction lasting >24 h, which is of a non-traumatic, vascular origin [1,2]. Although there is debate about the tissue- or time-based definitions of strokes, we chose to follow the time-based approach. This trend is expected to escalate in the future due to the aging of the population, resulting in a substantial increase in the number of stroke survivors and the associated financial burden [2]. Approximately 88% of all strokes are caused due to cerebral ischemia, with the remaining 10% and 2% attributable to intracerebral and subarachnoid hemorrhage, respectively. Among ischemic strokes, embolic etiology predominates, with approximately 35% categorized as cardioembolic, 45% as cryptogenic, and 20% attributed to atherosclerosis affecting major feeding vessels, notably the carotid arteries [3,4].

In current clinical guidelines, the grade of carotid artery stenosis is the basis of indication for carotid revascularization therapy [2,5,6]. Invasive intervention should be considered for average-surgical-risk asymptomatic patients with 60% to 99% stenosis, verified with one or more imaging results or based on clinical characteristics. The ESVS (European Society of Vascular Surgery) guideline listed that high risk for stroke may be associated with silent ischemic lesion on CT or MRI of the skull, progression of stenosis, large plaque area, large echogenic plaque, echolucent plaque on color duplex ultrasound (DUS), intraplaque hemorrhage (IPH) confirmed on MRI, reduced cerebrovascular reserve capacity, and spontaneous embolization on transcranial Doppler (TCD) [6]. However, none of these directions are recommended as part of the clinical routine diagnostic evaluation.

The latest and most detailed guideline on carotid artery stenosis treatment mentions the lack of a validated algorithm as a limiting factor for identifying asymptomatic patients with ‘high risk for stroke’ in whom invasive carotid artery reconstruction should be targeted [2,6]. There is no guideline or consensus on which imaging modality has overall priority in the assessment of plaque morphology and composition. Currently, the available body of evidence comprises findings predominantly drawn from a limited subset of case–control clinical trials. The accumulation of data derived from a broader patient cohort, essential for establishing routine clinical applicability, remains pending [7,8,9,10,11,12,13,14,15,16]. Based on the latest, most comprehensive review of the diagnostic imaging studies on high-risk vulnerable plaques published in 2020, the most common morphologies are neovascularization, echogenic plaques, and lipid-rich necrotic core (LRNC) [17]. Vulnerable plaques can be responsible for symptomatic or silent ischemic lesions due to embolization. Plaque morphology-based risk stratification might reduce the overtreatment of patients with high-grade stenosis alone. It potentially identifies patients with high-risk plaque but relatively lower degrees of stenosis.

The definition of perioperative stroke is the following: any embolic, thrombotic, or hemorrhagic cerebrovascular event with motor, sensory, or cognitive dysfunction lasting at least 24 h, occurring intraoperatively or within 30 days after surgery [18]. However, the risk of perioperative neurological events most frequently arises from diminished cerebrovascular reserve capacity, typically resulting in hypoperfusion-induced ischemia or procedural factors, rather than embolization linked to plaque-related mechanisms [19]. In our review, we focus on the risk assessment of perioperative neurological events caused by impaired cerebrovascular reserve capacity in patients with asymptomatic carotid artery stenosis.

No accurate risk estimate can be given with data obtained by only Circle of Willis (CoW) morphology, and no study has been undertaken to examine and evaluate the results of non-invasive peri- or intraoperative hemodynamic tests in a complex manner. Furthermore, there is a previously unanswered question listed in the ESVS guideline in 2023, which asks if the 3% threshold of 30-day stroke risk when performing carotid intervention should be reduced. The latest expert-based Delphi Consensus document suggests reducing this threshold to 2% [20]. To reduce the limit, there is a need for clear recommendations on non-invasive diagnostic tools in a forthcoming guideline, which could help to mitigate perioperative stroke risk by aiding optimal patient selection and providing more sensitive prediction of intraoperative hemodynamic changes during carotid clamping to avoid the immediate onset of neurological events or late onset of changes in cognitive functions.

The scope of this review is to summarize non-invasive pre- and intraoperative diagnostic methods that can supplement usually pre-existing CTA or MRA imaging. Through the identification of certain CoW variations associated with elevated risk, indications for further evaluation of cerebrovascular reserve capacity using additional diagnostic modalities can be delineated. A combination of CoW morphology assessment with these tools may offer a more accurate perioperative stroke risk estimation. The more accurate indication of carotid artery stenosis reconstruction and increased vigilance for selective shunt use during surgery could reduce the occurrence of perioperative ischemic neurological events and cognitive function decline due to hypoperfusion.

## 2. Pre- and Perioperative Risk Assessment of Impaired Cerebrovascular Reserve Capacity

According to the latest guidelines from the ESVS, DUS stands out as the primary imaging tool for diagnosing carotid artery stenosis. However, in cases where carotid endarterectomy is being considered, it is advised to confirm the duplex ultrasound findings with either computed tomographic angiography (CTA) or magnetic resonance angiography (MRA) [2,5]. These alternative modalities offer the advantage of comprehensive imaging, including visualization of the aortic arch, supra-aortic trunks, carotid bifurcation, distal internal carotid artery (ICA), and intracranial circulation. Notably, CTA and MRA can assess CoW morphology without additional imaging in the clinical diagnostic algorithm. However, there are no other routinely recommended imaging modalities capable of providing information on impaired cerebrovascular reserve capacity in daily clinical practice. An intriguing question is whether variations of CoW morphology could suggest the need for further non-invasive diagnostic evaluation in the clinical diagnostic algorithm to more precisely predict perioperative neurological events or the risk of cognitive decline.

### 2.1. Carotid-Related Cerebral Perfusion via the Collateral Vascular System

Cerebrovascular circulation relies on four major feeding vessels. The two common carotid arteries (CCAs) originate from the aortic arch. The right CCA has a common origin with the right subclavian artery, called the innominate artery. The CCAs divide into the internal (ICA) and external carotid arteries (ECA). The ICA then runs through the carotid canal of the temporal bone to the base of the skull to join the Circle of Willis (CoW). Before joining the CoW, the ICA gives off its sole intracranial branch, the ophthalmic artery. Conversely, the posterior cerebral circulation primarily relies on the vertebral arteries, arising from the subclavian arteries and converging to form the basilar artery, eventually joining the CoW. Cerebral collaterals are vascular redundancies in the cerebral circulation that can partially maintain blood flow to ischemic tissue when primary conduits are blocked. Blood vessels can form anastomoses or connections that could potentially protect the brain from infarction or limit the amount of damage by providing alternative routes for blood to reach brain regions threatened with ischemia. The term ‘collaterals’ will be used for any kind of cerebrovascular anastomosis and connection in this review. The collateral circulation of the brain can be divided into three functional anatomical groups. On the base of the brain, connecting all major feeding vessels is the Circle of Willis, which is considered the primary collateral system. Microvascular intracranial collaterals consisting of leptomeningeal and subcortical collaterals make the secondary collateral system with the addition of extracranial–intracranial vascular anastomoses as the third system. These collaterals consist of the following anastomoses: connection between branches of the facial artery and ophthalmic artery, supraorbital and supratrochlear arteries and the ophthalmic artery, the occipital artery and the posterior cerebral arteries (PCAs), the middle meningeal artery and the anterior cerebral arteries (ACAs), and finally branches from the cervical arteries to the vertebral artery [21]. 

#### 2.1.1. Circle of Willis

The CoW is considered the most important collateral network that connects the major feeding arteries of cerebral circulation. The CoW can be divided into functional anatomical segments. Two posterior arches connect the basilar artery to the middle cerebral arteries (MCAs), consisting of the first segment of the posterior cerebral artery (PCA) and the posterior communicating artery (Pcom). The anterior arches of the CoW consist of the first segment of the anterior cerebral arteries (ACAs) and the anterior communicating artery (Acom).

##### Epidemiology of CoW Variabilities

A high variability in the CoW has been published [22]. However, it is difficult to compare various studies, because of the different criteria and methods used to define the complete or compromised state of the circle; this possible bias for comparison is summarized in Table 1. The variations in CoW segment classification outlined in Table 1 underscore the critical need for standardization in research methodologies, as these differences significantly impact the comparability and reliability of the findings of various studies; therefore, the definition or non-definition of hypoplastic segments can change the propensity of complete CoW. Prevalence of complete CoW ranges from 13% to 59% in postmortem studies and 12 to 79% in imaging studies [23,24,25,26,27,28,29,30,31]. Variations tend to be more common among patients with cerebrovascular disease as opposed to control patients [32,33]. The incidence of complete CoW is likely greater in women for all age groups and likely decreases with age in both genders [34,35].

##### Effects of CoW Variations on Cerebrovascular Hemodynamics and Stroke

According to model and in vivo studies, anterior circulation plays a greater role than posterior circulation during acute unilateral carotid occlusion. The ipsilateral carotid occlusion with a contralateral aplastic ACA presents the worst scenario with a reduction in ipsilateral MCA flow rate by almost 40% [41]. A dominant contralateral ACA and a patent anterior communicating artery are required to conduct flow from the opposite hemisphere. In patients with high-grade contralateral ICA stenosis, the posterior circulation has been found to have a greater influence as reported by an in vivo study [42].

A recent meta-analysis found that the presence of any variation (incompleteness or hypoplasia) in CoW increases the risk of ischemic stroke; however, the result was not statistically significant [OR: 1.38 (95% CI 0.87, 2.19)] [43]. Recent studies imply that incompleteness of CoW not only contributes to the development of ischemic stroke but can worsen its outcome. Retrospective analysis of stroke patients’ MRA scans suggests that having an incomplete CoW can decrease the chances of the patient having a good clinical outcome in the short term by 47% (*p* = 0.046, OR 0.53, 95% CI 0.281–0.988) [37]. The different criteria used to define a complete or incomplete CoW, as well as the differing cohort size and imaging modality (CTA or MRA) used in each study may account for the variability in these data.

##### Effects of CoW Variations on Intraoperative Ischemic Stroke Risk

CoW shows great variability, and these variations seem to influence cerebral circulation during carotid reconstruction, especially during clamping. Analyzing CoW variations can help determine planned shunting strategies before carotid endarterectomy (CEA).

Intriguing questions arise regarding the impact of carotid artery occlusion on blood flow within the CoW and whether it is capable of providing compensation for this occlusion to avoid the development of stroke in patients. A Canadian research group found that during temporary carotid artery occlusion, carotid stump pressure (SP) was influenced by mean arterial blood pressure, contralateral carotid artery diameter, and the flow from the contralateral carotid, indicating that the posterior part of the Circle of Willis maintains blood flow during clamping in high-grade contralateral carotid stenosis or occlusion, while the role of the anterior semicircle becomes more crucial with sufficient flow from the contralateral carotid artery [42]. In patients undergoing carotid endarterectomy, certain variations in the CoW, especially those that lead to the isolation of the MCA, significantly elevated the risk of immediate neurological events (OR 11.12; 95% CI, 3.57–35.87; *p* < 0.001). Preoperative assessment of the capability of the CoW to form a patent collateral pathway may identify patients requiring shunt protection during clamping [44]. In another prospective case series, non-functional CoW collaterals were identified as a risk factor for ischemic events following carotid cross-clamping during carotid endarterectomy. Patients experiencing post-cross-clamping ischemic symptoms exhibited absent or hypoplastic CoW segments and a notable decrease in ipsilateral brain tissue oxygen saturation [36].

#### 2.1.2. Role of the Secondary Cerebral Collaterals

Other collateral channels and anastomotic systems such as leptomeningeal collaterals and extracranial to intracranial anastomoses, for example, with retrograde flow in the ophthalmic artery, are defined as secondary cerebral collateral blood flow. Reversal of blood flow in the ophthalmic artery can provide additional support in individuals with carotid artery stenosis, facilitated by connections to various arteries through the orbital plexus, potentially improving hemodynamics in chronic cerebral hypoperfusion [45,46]. Further cortical anastomoses link the terminal branches of each cerebral artery at the surface of the cortex and into the leptomeningeal area, where the pial arteries penetrate the gray matter. An anastomotic collateral supply system can develop to link the external carotid artery and the ICA as well.

Several factors affect the functionality of these collaterals. It has been demonstrated that aging adversely affects the adequacy of leptomeningeal collaterals, leading to more extensive tissue loss in ischemic stroke [47]. Factors associated with the development of atherosclerosis also seem to have an impact on cerebrovascular capacity. Untreated chronic hypertension was reported to cause vasoconstriction in pial collaterals. Diabetes has also been associated with diminished flow in an animal model [48]. The adverse effects of metabolic syndrome and old age on cerebral collateralization have been suggested by Menon et al. in a study conducted on over 200 patients with ischemic stroke [49]. The effects of an ischemic precondition are controversial in the cerebral circulation, although a retrospective assessment of 600 patients who suffered an ischemic stroke has found a positive correlation between current or prior smoking and good collaterals. The beneficial effects of regular physical exercise and statin use have been reported on collateral formation [50,51].

These modifiable risk factors can be taken into account when recovery from ischemic stroke is estimated and can also serve as warning signs in patients undergoing procedures that temporarily disturb regular cerebral perfusion, for example, clamping during carotid reconstruction. Several systemic factors (decreased renal function, hypertension, and carotid occlusion) have a significant negative effect on ocular microcirculation [52]. Decreased cerebrovascular reserve capacity has been strongly associated with elevated stroke risk in patients with significant internal carotid artery stenosis [53].

### 2.2. Non-Invasive Diagnostic Tools to Measure and Evaluate Cerebrovascular Reactivity

Cerebrovascular reactivity (CVR) is the mechanism behind the process which occurs through the constriction and dilation of cerebral resistance vessels to maintain a constant blood flow. While static autoregulation assesses long-term changes in blood pressure, dynamic autoregulation focuses on immediate changes within seconds to minutes [54]. The latter is particularly relevant in scenarios such as carotid artery reconstruction with carotid clamping without shunting, where the timing of CVR changes is critical. In the upcoming text, we will delve into various non-invasive diagnostic tools, such as imaging of retinal circulation by optical coherence tomography angiography (OCTA) and classical non-invasive neuromonitoring methods, such as functional transcranial Doppler (fTCD) and near-infrared spectroscopy (NIRS).

#### 2.2.1. Imaging of Retinal Circulation

The retinal microvasculature offers a unique opportunity to investigate the pathogenesis of cerebral small vessel disease. The reason for this is that the cerebral and retinal circulations share similar anatomy, physiology, and embryology [55]. Similarly to cerebral microcirculation, the retinal system lacks anastomoses, acts as a barrier with autoregulatory capabilities, and functions as a relatively low-flow, high-oxygen extraction system [56]. Through directly visualizing retinal vessels, we can investigate retinal microvascular status as an indicator of brain microvascular health. The retina receives its blood supply from the retinal and choroidal vasculature, both of which originate from the ophthalmic artery, the first intracranial branch of the ICA. Therefore, deviation in blood flow parameters in the ICA has the potential to induce ophthalmic complications. Pathophysiological changes affecting the central nervous system and cerebral microcirculation can influence the retina and retinal microcirculation owing to shared cellular, molecular, thromboembolic, and hemodynamic mechanisms. Several methods are available for imaging ocular circulation, such as fundoscopy and fluorescein angiography, but these diagnostic tools do not allow for the quantitative measurement of blood supply. Doppler sonography of ocular arteries offers a richer insight into estimating blood supply than conventional angiography. However, it is limited to large ophthalmic arteries and is therefore less informative for studying retinal microcirculation [55].

##### Optical Coherence Tomography Angiography

More recent methods can provide more accurate functional non-invasive measurements. Optical coherence tomography (OCT) can produce high-resolution, cross-sectional images of the retina and choroid. OCTA is a novel method of imaging and analysis of the retinal and choroidal vasculature, which does not require the use of intravenous dye. Utilizing motion contrast technology, this imaging modality is capable of detecting red blood cell movement within vessels, as well as of precise visualization of the microvasculature in the macular area and around the optic disc. Additionally, it furnishes quantitative information, such as vessel density in the above-mentioned areas and the size of the foveal avascular zone. The examination is rapid, easily repeatable, and provides simultaneous structural and functional blood flow information.

However, this methodology is not without its constraints. Capillaries with blood cell velocities below the detectable threshold cannot be visualized, as in the case of bleeding and leakage. Currently used OCT devices and software only allow measurement of the central parts of the retina: a 3–8 mm square of the macular area or a 4.5–6 mm square of the peripapillary area. Predominant limiting factors for OCTA include image quality issues, such as media opacities and artifacts resulting from blinking or saccadic eye movements. The device automatically calculates the signal quality of each scan yielding a signal quality score. Previous studies have emphasized the significance of image quality, as it affects measurement error, which is significantly greater in scans with lower-quality scores [57,58,59,60,61,62]. Despite being a non-invasive procedure, OCTA is quickly becoming more prevalent in clinical routine. For instance, it is being used to detect early microvascular alterations in diabetic retinopathy, visualize the boundaries of nonperfusion in vascular occlusion, and monitor patients with age-related macular degeneration. Additionally, OCTA shows promise as a diagnostic and monitoring tool for glaucoma and carotid artery stenosis (CAS) [55], even if it is not well established in the vascular surgery preoperative diagnostic protocols.

Patients with CAS (similarly to other cardiovascular systemic diseases) have, in general, lower retinal capillary vessel density [63]. Retinal flow density values measured with OCTA and stimulation-evoked retinal venous dilation are reduced in CAS patients compared to healthy controls [64,65]. Furthermore, OCTA can detect minor changes caused by mild CAS [66]. After carotid reconstruction, substantial enhancement in the retinal capillary network and vessel density in the macular deep vessel complex in both eyes compared to healthy subjects was observed by OCTA (Figure 1B) [64,67].

There are a few publications on animal models of retinal ischemic injury by unilateral and bilateral common carotid occlusion with evidence of retinal ischemia by blocking collateral flow from the CoW [68,69]. Previous intraoperative measurements have suggested that the patency of the Circle of Willis can be confirmed by monitoring the orbital vessels. These findings suggest that retinal blood flow may provide an “acoustic window” into intracranial circulation during cardiac surgery [70]. A recent study found that patients with an uncompromised CoW had better ocular perfusion than those with a compromised CoW [71]. The same study also found a notable enhancement in retinal capillary perfusion in both eyes after CEA, suggesting that in the event of carotid artery occlusion, patients with a non-compromised CoW exhibit better-preserved retinal blood flow than subjects with a compromised CoW owing to the remodeling of the intra-orbital blood flow.

#### 2.2.2. Monitoring of Cerebrovascular Circulation

Transcranial Doppler (TCD), introduced by R. Aaslid in 1982, offers real-time monitoring of blood flow velocity in major cerebral arteries through different insonation windows of the cranium. These are the transtemporal, transorbital, foraminal, and transmandibular windows [72]. TCD holds significant clinical value due to its non-invasive nature, ease of use, and high temporal sensitivity, making it a widely employed tool in routine clinical settings for diagnosing conditions such as vasospasm after subarachnoid hemorrhage, assessing signs of global cerebral ischemia when brain death is suspected, detecting microemboli (in a laboratory setting or even on real time, during carotid surgery or interventions), and evaluating cerebral vasoreactivity to vasoactive stimuli [73].

##### Cerebrovascular Reserve Capacity Estimation by Functional Transcranial Doppler

In recent decades, several studies have been published discussing reduced cerebrovascular reactivity (CVR) as a relevant ischemic risk factor in patients with severe ICA stenosis [74,75]. Among the many validated vasoactive stimuli, tests based on CO_2_ reactivity (e.g., breath-holding test, BH), the Valsalva Maneuver (VM), and the common carotid artery compression (CCC) test are used the most commonly. The most notable limitation of the previously presented tests, as assessed by using functional TCD (fTCD), is the poor insonation window; therefore, examination of older populations with significant ICA stenosis can sometimes be challenging [76].

Among the CO_2_ reactivity tests, the BH test is considered to be the most physiological, as it is minimally burdensome for the tested person and it does not represent an invasive intervention or an unpleasant experience, in contrast to the acetazolamide test using intravenous injection or CO_2_ inhalation, which in many cases causes a feeling of suffocation [77]. Based on the result of the BH test, the breath-holding index (BHI) can be calculated, which is related to the sensitivity of the CVR, although it does not characterize its capacity. According to clinical experience, a BHI lower than 0.69 indicates an impaired CVR [78].

VM can also be considered a physiological stimulus, during which the subject increases the intrathoracic pressure by forced exhalation, thereby causing significant changes in arterial blood pressure and, consequently, in the velocity of cerebral blood flow in the proximal cerebral arteries. Elderly patients can also perform the maneuver excellently. The characteristic four phases of the reaction can be easily identified in most of the measurements. At the beginning of the maneuver, the intrathoracic pressure is temporarily increased, which is then transferred to the arterial system, causing the blood pressure (ABP) to rise. The increased intrathoracic pressure limits the normal venous return, and the atrioventricular filling decreases, causing a temporary decrease in ABP. This phase is marked as “IIa”. Stage IIb is the phase of compensation; during this stage, the reduced systemic ABP is felt by the baroreceptors and is followed by an increase in sympathetic tone in the periphery as a response. Phase III marks the end of the maneuver, at which point the blood pressure temporarily decreases due to the sudden drop in intrathoracic pressure but is followed by a temporary increase (Phase IV) due to the previously activated sympathetic tone and the normalizing atrioventricular filling. The maximum point of phase IV is called the overshoot reaction (OS). When the reaction subsides, reflex bradycardia occurs. Parallel to the BP changes, simultaneous reactions of the cerebral vessels can be identified. The increased intrathoracic pressure leads to a significant decrease in arterial blood pressure by obstructing the venous filling of the heart, which simultaneously activates the sympathetic cardiovascular system and cerebral pressure autoregulation, causing vasoconstriction in the periphery and vasodilation in the brain [79,80,81]. In order to evaluate these functions, several parameters have been defined earlier such as the Autoregulation Index (ARI), the Cerebrovascular Valsalva Ratio (CVAR), the Peripheral Valsalva Ratio (PVR), and the Critical Closing Pressure (CrCP). For testing the integrity of the cardiovascular autonomic functions, the Valsalva Heart Rate Ratio (VHRR), the pressure recovery time (PRT), and the sympathetic index (SI) are measured [82,83,84,85].

In patients with impaired CVR and autonomic integrity, several deviations can be observed compared to a healthy VM reaction. These include the lack of increase in BP and heart rate in the second half of the compression phase and in the last, recovery phase after the end of the expiration effort. Significant cardiovascular autonomic dysfunction can be verified most often in patients with diabetes mellitus, in case of some neurodegenerative diseases (multiple system atrophy, Parkinson’s disease, progressive supranuclear palsy), as well as in some patients with severe atherosclerotic carotid stenosis. In the latter cases, the risk of cerebral ischemia increases [86,87]. 

The CCC test is also suitable for estimating cerebral vasoreactivity, and it also models the effect of carotid clamping used during reconstruction surgery. Importantly, this method requires a prior carotid artery ultrasound examination to reduce the risk of provoked embolization. During the examination, manual compression is performed on the common carotid artery (CCA), which causes a temporary decrease in blood flow velocity (BFV) of the ipsilateral MCA. Based on previous research, the CCC test has a preoperative predictive value of 66.7% for a positive outcome and 100% for a negative outcome. It also has a sensitivity of 100% and a specificity of 97.8% in predicting the need for a shunt during carotid endarterectomy [88,89,90,91,92]. In a larger study encompassing a wider cohort of patients with CAS, the CCC test was found to have a 60% likelihood of indicating the need for intraoperative shunt use [91,92]. To enhance this indication, the CCC test can be combined with other functional TCD tests [91,92]. Based on previous experience, the recommended duration of compression is 10 s, which is sufficient to induce maximal reactive hyperemia, called transient hyperemic response (THR). The degree and dynamics of THR reflect the latency and effectiveness of cerebral vasoreactivity. Based on BFV values during CCC, the transient hyperemic response ratio (THRR) can be calculated (Figure 1C). Patients with significant ICA stenosis often have a delayed response after the release of the compression on the side of the stenosis [89,90]. Recent studies have defined new variables that also predict other aspects of the impairment of the CVR—the delayed THRR (DHTRR) and prolonged return to baseline (RTB) time. According to a single-center prospective study, compromised CoW in CAS patients was associated with altered cerebral hemodynamics during CCC test evaluation. This alteration manifested as reduced blood flow velocity, reduced transient hyperemic response, and increased cerebrovascular resistance compared to controls and CAS patients with an intact CoW [90].

It is important to note as a limitation that both fTCD and CCC require skilled operators to perform and interpret the tests accurately. Also, fTCD measures flow in large arteries, which may not fully reflect regional flow variations in smaller vessels. The measurement of blood flow velocity and its change in proximal cerebral vessels by TCD without knowledge of the arterial pressure, which maintains perfusion in itself, does not reflect the change in blood flow in the relevant brain tissue. Furthermore, patient safety needs to be considered when using CCC due to the risk of embolization caused by the manual compression of CCA. Overall, the presented functional tests above are all validated stimuli that are considered feasible to evaluate CVR. Parallel to the assessment of the complex cerebrovascular reactions, the tests provide further information on cardiovascular autonomic integrity. By using the listed reactivity tests together, the ischemic risk of carotid patients can be determined more precisely, which makes it easier to decide on the indication for surgery and choose the appropriate surgical technique.

##### Non-Invasive Neuromonitoring by Near-Infrared Spectroscopy

NIRS enables ongoing, immediate observation of regional cerebral oxygen saturation (rSO_2_) within the frontal cortex by analyzing reflected near-infrared light. This method furnishes data on the average oxygen saturation of hemoglobin across arterial, capillary, and venous blood, indicating the oxygen levels remaining post-extraction by tissues and essential organs. Numerous factors can affect the recorded rSO_2_ values, such as arterial oxygen saturation, systemic blood pressure, arterial carbon dioxide levels, hematocrit levels, cerebral blood volume, and variations in baseline rSO_2_ levels among individuals and over time [93]. NIRS can provide indirect, but real-time information about the blood flow of the frontal cortex. The changes and tendencies of the rSO_2_ values are more informative than the absolute values.

During carotid artery clamping, rSO_2_ monitored on the ipsilateral forehead drops as ipsilateral cerebral perfusion and oxygen delivery decreases (Figure 1D) [94,95]. The rSO_2_ reduction was shown to correlate with changes in electroencephalography (EEG), TCD, stump pressure (SP), and postoperative neurologic deficits [94,95,96,97,98]. Several studies showed varying sensitivity (30–100%) and specificity (77–98%) for detecting cerebral ischemia using NIRS in CEA patients under locoregional anesthesia (LA) and general anesthesia (GA) [95,98,99,100,101]. According to a meta-analysis, studies quantifying the diagnostic accuracy of NIRS under GA were much more variable than under LA due to the impact of the anesthetic technique [102]. At present, there is no consensus on a critical rSO_2_ threshold, below which cerebral ischemia may develop and shunting is warranted. However, in previously published studies, patients with a relative drop in rSO_2_ ranging from 9% to 25% were indicated for shunting during carotid endarterectomy [94,95,96,97,98,99,100,101]. The latest systematic review discussed that NIRS monitoring using a 20% threshold is specific but not sensitive, leading to low rates of unnecessary shunting at the expense of missed identification of ischemic events in LA [102].

The relative change in rSO_2_% is primarily determined by comparing measured rSO_2_ values to baseline rSO_2_ levels. Variations in how baseline rSO_2_ is calculated across studies can significantly impact calculated relative rSO_2_% values. Previous studies have examined a two-minute test clamping period to indicate shunting severity based on rSO_2_% decline, but evidence suggests that this may not accurately reflect rSO_2_% changes throughout the entire clamping period [101,103]. Additionally, the use of different formulas to calculate relative rSO_2_% change and the lack of a standardized formula for baseline and rSO_2_% calculation further complicate comparisons between studies. Moreover, the availability of various commercially available NIRS devices, each with unique properties, can lead to inconsistent rSO_2_ measurements even within the same individual due to calibration disparities.

### 2.3. Biomedical Modeling of Cerebrovascular Hemodynamics in Patients with CAS

Stenoses significantly affect the flow locally at the carotid bifurcation and have an effect further downstream in the arterial system. Therefore, numerous engineering and mathematical studies have developed models to assess the flow. Computational fluid dynamics (CFD) is a widely used tool in biomedical research. The biomedical models concentrate either on the flow locally around the bifurcation, resolving the blood flow in a detailed three-dimensional flow simulation, or use one-dimensional simulations to determine the flow globally on the system level. One of the pioneering works on the topic of a 1D mathematical model of arteries was Avolio’s in 1980, but numerous papers have been published on the topic since then [104,105,106,107]. There are biomedical models of the human arterial system that can be made patient-specific to assess the effect of carotid cross-clamping [108].

#### 2.3.1. Effect of CoW Variations on Hemodynamical Modeling

Since the impaired vasoreactivity concerning the CoW has a significant effect on the risk of ischemic stroke and its outcome, biomedical studies often deal with the simulation of the flow in the CoW. The simplest models are based on the mathematical formula of the Hagen–Poiseuille principle or the continuity equation of fluids [109,110]. These studies concentrate mainly on the flow inside the CoW. However, these models can be augmented with the occlusion in the ICA to assess the role of the CoW in stenotic ICAs [111,112].

A significant part of the population has an incomplete CoW; therefore, the mathematical flow models often incorporate the effects of CoW variations [41,113]. These models show that cerebral blood flow remains in a physiologically acceptable range; however, the flow inside the segments of the CoW greatly changes because of the variations, which may result in an unfavorable hemodynamic load (Figure 1A). Alastruey et al. also discuss CoW variations, which can be critical in the case of an ICA occlusion [41].

Apart from one-dimensional simulations, the CoW can be modeled in 3D as well. Therefore, studies concentrating on the CoW variations use 3D flow simulations [114,115]. An automatic software for the segmentation of CoW morphology and topology based on CTA and MR imaging will be available in the daily clinical routine within a couple of years [116]. The automatic segmentation software will be able to assist and reduce the bias in engineers’ or clinicians’ segmentation preparation for further research on CoW modeling. The simulations show the flow reserve of the CoW in the case of serious ICA occlusion, similarly to the 1D models. The other result of such simulations can be the investigation of emboli, which travel in the blood to different locations in the brain. The CoW variations have a significant effect on the aggregation of such emboli.

#### 2.3.2. Hemodynamic Effects of Carotid Artery Stenosis

The local hemodynamic environment of carotid stenosis has been of great interest to the computational hemodynamic community. However, a proper simulation for stenosis requires adequate modeling strategies on multiple levels. First, the computational model of lumen volume is challenging to segment out from medical images. Accordingly, the geometry creation has the largest uncertainty on the outcome of CFD simulations; hence, proper care is mandatory at this step [117]. Secondly, the applied physiological conditions proposed as boundary conditions also have a substantial impact on the simulations [118,119,120].

Using CFD, investigations showed that an unsteady wall shear stress (WSS) exposure can be found even in a healthy carotid bifurcation [121,122]. Others demonstrate that even transitional flows can occur after the stenotic vessel section [123,124]. The focus of the research is the quantification of this unsteady flow field, mainly based on the WSS field, as it is believed that it can mitigate the mechanotransduction process [125]. One of the next great challenges in the field is to connect computational hemodynamics and plaque analysis, maybe with corresponding histopathological investigations [126].

## 3. Cognitive Function Changes in Patients with Carotid Artery Stenosis

Intact cognition is essential for an independent life of good quality, including not only the patient but their whole family. It is well-known that impaired postoperative cognitive function increases the duration of hospital stay, the need for rehabilitative services, healthcare utilization, as well as long-term mortality, and also impairs mobility [127,128,129]. Individuals experiencing cognitive impairment face an increased likelihood of prematurely exiting the workforce, consequently becoming reliant on social welfare benefits [130]. Besides the above-mentioned negative consequences, cognitive impairment means a major financial burden not only for the patients and their families but also for the whole economy [131,132].

Baseline cognition of asymptomatic patients with severe carotid stenosis showed below-normal cognition compared to the population-based cohort in CREST-2 study preliminary results [133]. Asymptomatic carotid stenosis with impaired cerebrovascular reserve is associated with cognitive decline [134]. According to a systematic review from 2021, patients with severe carotid artery stenosis and impaired CVR are more likely to have cognitive impairment and to suffer further cognitive decline with time [135]. The latest systematic review suggests that there is an association between asymptomatic carotid stenosis and progressive cognitive deterioration due to cerebral hypoperfusion and silent embolization [136,137,138].

### 3.1. Prevention of Further Cognitive Decline with Carotid Reconstruction

The effect of carotid endarterectomies on postoperative changes in cognitive function is a frequently investigated topic, without obvious consequences [2,137,138]. Theoretically, removing the plaque, and therefore optimizing cerebral blood flow and eliminating potential embolic sources, should improve cognitive function. However, in everyday clinical practice, the benefits of carotid procedures regarding cognitive function are not so obvious [129,138]. The explanation of these controversial results analyzing the impact of carotid endarterectomy on postoperative cognitive function may be the diversity of the studies, the almost complete lack of randomized controlled trials, the definition of control groups, no protocol for pre- and postoperative cognitive test assessment (timing, exact tests, more specific and standardized neuropsychological tests), and the complex interaction of factors affecting postoperative cognitive function. The potentially deleterious effects of anesthesia, cerebral hyperperfusion, microembolization, and intraoperative cerebral hypoperfusion further complicate the issue [137]. Among these, microembolization and cerebral perfusion seem to be the most relevant factors, although their effect on postoperative cognitive function remains uncertain as well [134,139].

Cerebral perfusion—the other key factor regarding postoperative cognitive function—might be monitored by TCD or SP, since the change in regional cerebral tissue saturation correlates with the change of cerebral hemodynamics [140]. Although the results of the studies analyzing the connection between regional cerebral tissue saturation (rSO_2_) by NIRS and postoperative cognitive function are not completely concordant, an overwhelming number of studies—though from other fields of medicine—have proven the clear connection between cerebral tissue saturation monitored by NIRS and postoperative cognitive function [140,141,142,143,144,145,146,147].

### 3.2. The Role of Cerebrovascular Reserve Capacity in Patients with Carotid Stenosis and Cognitive Dysfunction

The relationship between decreased cerebrovascular reserve capacity and diminished cognitive function in specific domains in patients with CAS was last researched more than a decade ago [148,149]. The controversial results for cognitive function affected by CAS could be explained in the context of variation in cerebral vasomotor reactivity among patients with asymptomatic stenosis based on a prospective study, which evaluated cognitive status for three years and compared it to cerebral hemodynamics, measured by fTCD with the breath-holding index test. Patients with impaired reactivity were more liable to have cognitive impairment [150]. Another clinical study of symptomatic carotid stenosis patients showed decreased values of CVR on the stenotic side by TCD and reduced scores on the cognitive tests, which was enhanced at 6 months after carotid artery reconstruction. They concluded that the improvement was directly related to the increase in the vasomotor response on the side of revascularization [151]. A systematic review summarized the results of five studies on decreased cerebrovascular reserve capacity measured by TCD with BH test and cognitive impairment. It concluded that CVR may serve as an additional tool to determine whether patients are in fact symptomatic from their carotid stenosis and should be considered for intervention [152].

There are no published results in the literature about OCTA measurements in CAS patients with longitudinal studies to follow up on cognitive function. Some results indicate that OCT findings can help identify the etiology of cognitive decline and/or serve as objective markers of Alzheimer’s disease [153]. Retinal vessel density measured by OCTA correlated with decreased CVR in a small-sample-size, cross-sectional study, and based on these results, cognitive dysfunction and vessel density may have a connection as well [89]. There is a need for further investigation into this topic.

Cognitive impairment is an underrated comorbidity in patients with ACS, which can be influenced if it is the result of impaired cerebral hemodynamics, which is a possible new indication for carotid artery reconstruction. Furthermore, the changes in cognitive function could serve as a useful, sensitive, and non-invasive tool to detect the long-term consequences of perioperative cerebral hypoperfusion as well.

### 3.3. Effects of the CoW on Cognitive Dysfunction in Patients with Carotid Stenosis

Besides the above-mentioned points, the anatomy of the CoW has a non-negligible impact on cognitive function. In a two-year follow-up study, the presence of robust collateral circulation in cases of severe middle cerebral artery stenosis correlated with a reduced risk of cognitive impairment, contrasting with patients exhibiting poor to moderate leptomeningeal collateral systems who experienced deterioration in cognitive domains [154]. This connection was confirmed by another study, analyzing the effect of cerebral artery stenosis of anterior circulation on cognition [155]. Furthermore, an association was found between poor collateralization within the CoW and early cognitive dysfunction after carotid endarterectomies. The status of the posterior communicating artery was an independent predictor for early cognitive dysfunction [156]. Another study demonstrated that an effective collateral flow may promote the normalization of the perfusion territory after carotid endarterectomies, which was associated with better postoperative cognitive outcomes [157]. It is a promising field for further research to compare the effect of CoW variations on CVR and their connection with cognitive function and its changes after revascularization.

## 4. Future Directions

Drawing upon the insights gleaned from this review, several prospective directions can be delineated. In the foreseeable future, the methodical integration of non-invasive diagnostic modalities such as optical coherence tomography angiography (OCTA), functional transcranial Doppler (fTCD), and near-infrared spectroscopy (NIRS) in both preoperative and intraoperative contexts holds substantial promise for refining clinical decision-making processes in patients with higher risk for perioperative cerebral ischemia. For patients exhibiting concerns regarding Circle of Willis (CoW) morphology in preoperative imaging, this holistic approach endeavors to attenuate the risk of perioperative ischemic neurological events and pre-empt subsequent cognitive decline, potentially culminating in superior outcomes for asymptomatic patients and could define a subgroup of patients for additional preoperative diagnostic assessment for further risk stratification.

However, an initial imperative lies in rectifying existing methodological biases. This necessitates the formulation of robust protocols and consensus materials for CoW segment classification, meticulous OCTA quality control protocols, standardization of fTCD measurements, and the elucidation or establishment of NIRS regional oxygen saturation (rSO_2_) baseline calculation and threshold for serious desaturation. Moreover, the establishment of standardized cognitive assessments tailored specifically to carotid disease is imperative. Following these standardization efforts, meticulously designed randomized clinical trials are essential for revalidating initial findings. Additionally, employing 1D and 3D blood flow models based on preoperative imaging offers the opportunity to assess each patient’s anatomical variations and pathological status before carotid reconstruction, with or without shunt replacement, under diverse hemodynamic conditions. Ultimately, the integration of these findings into a comprehensive clinical flowchart for guideline development holds promise for enhancing risk stratification by incorporating pertinent characteristics of high-risk patient cohorts, derived from clinical characteristics and plaque assessments, thereby reducing the overall risk of perioperative cerebral ischemia. The issuance of guideline-level recommendations for intraoperative shunting, informed by non-invasive preoperative additional imaging, hemodynamic modeling, or intraoperative NIRS desaturation levels, could significantly contribute to achieving the targeted reduction of the 30-day stroke risk threshold to 2%, as outlined in the latest Delphi consensus paper [20].

## 5. Conclusions

The preoperative identification of potential hypoperfusion tendencies resulting from compromised CoW and secondary collateral systems could influence treatment decisions aimed at preventing ischemic neurological events and mitigating the risk of cognitive decline progression. Detection of a compromised CoW in preoperative routine diagnostic imaging could be an indication for further non-invasive tests to measure the impairment of cerebrovascular reserve capacity and to offer perioperative risk stratification and management. Strategically employing intraoperative shunting based on preoperative predictions and real-time measurements may reduce the risk of perioperative hypoperfusion-related neurological events or delayed cognitive changes. Intraoperative neuromonitoring should aim to predict hemodynamic instability during surgery and provide opportunities for correction.

However, further targeted complex clinical studies are required to establish diagnostic algorithms and precise thresholds based on universal clinical protocols and standardized neuropsychological tests. Despite promising initial findings, robust data from randomized clinical trials or well-designed longitudinal studies are necessary to validate these new insights.

## Figures and Tables

**Figure 1 jcm-13-02487-f001:**
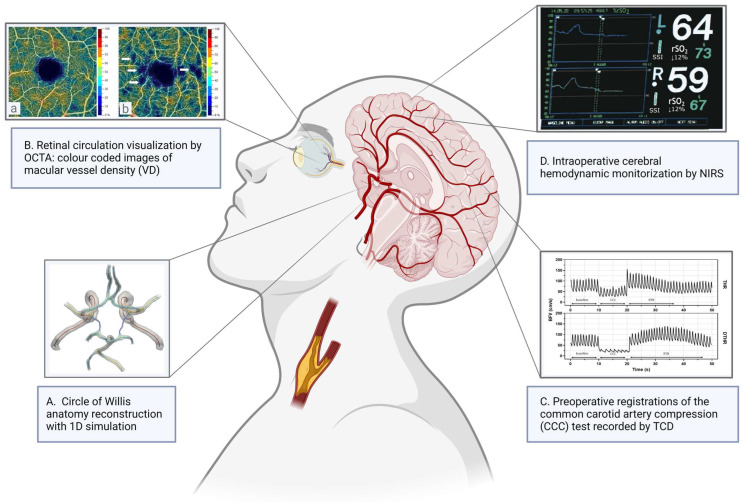
Schematic overview of preoperative and intraoperative, advanced, non-invasive diagnostic tools for the consequences of internal carotid artery (ICA) stenosis. (**A**) Circle of Willis (CoW) anatomy reconstructed by 1D simulation. (**B**) Optical coherence tomography angiography (OCTA) provided color-coded images of macular vessel density (VD) in a young, healthy volunteer (a; VD: 45.9%) and a patient with significant internal carotid artery stenosis (b; VD: 41.3%). (**C**) Transcranial Doppler (TCD) registrations of the common carotid artery compression (CCC) test and calculated parameters for the estimation of cerebrovascular reactivity. DTHRR = delayed transient hyperemic response ratio; RTB = return to baseline time. (**D**) Intraoperative monitoring of cerebral hemodynamic changes during carotid endarterectomy by near-infrared spectroscopy (NIRS). NIRS monitoring shows a typical curve of a patient with shunt placement. Created with www.BioRender.com. Accessed on 23 April 2024.

**Table 1 jcm-13-02487-t001:** Examples of different criteria used for determining variations in Circle of Willis (CoW) segments and the assessed completeness of the circle in imaging and postmortem studies.

Author	Year	Study Type	Criteria for Aplastic or Missing CoW Segment	Criteria for Hypoplastic or Stenotic CoW Segment	Complete CoW (%)
Gyöngyösi et al. [36]	2023	Imaging (CTA)	Non-visible	<0.5 mm	71.2%
Hindenes et al. [31]	2020	Imaging (MR-3D TOF)	<1 mm	Not evaluated	11.9%
Lin et al. [37]	2023	Imaging (MR-3D TOF)	Vessel diameter less than 50% of the contralateral or ipsilateral side (other segments)	Not evaluated	66.0%
Wholey et al. [38]	2009	Imaging (CTA)	Non-visible	Not evaluated	18.0%
Hartkamp et al. [25]	1999	Imaging (MR-3D TOF)	Non-visible	<0.8 mm	63% patients, 47% controls
Karatas et al. [39]	2015	Imaging (CTA)	Non-visible	<1 mm	71.0%
Li et al. [28]	2010	Imaging (CTA)	Non-visible	<1 mm	27.0%
Krabbe-Hartkamp et al. [27]	1998	Imaging (MR-3D TOF)	Non-visible	<0.8 mm	42.0%
Waaijer et al. [40]	2007	Imaging (CTA)	Non-visible	<1 mm	22.0%
Puchades-Orts et al. [29]	1975	Autopsy study	Non-visible	Not defined	13.0%
Riggs and Rupp [30]	1962	Autopsy study	Non-visible	Not defined	21.0%
Alpers et al. [24]	1958	Autopsy study	Non-visible	<1 mm	52.3%
Alpers and Berry [23]	1963	Autopsy study	Non-visible	Not defined	52% in normal brains, 33% in pathologic brains

## Data Availability

Not applicable.

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
