# Peer review of "Non-Invasive Tools in Perioperative Stroke Risk Assessment for Asymptomatic Carotid Artery Stenosis with a Focus on the Circle of Willis"

_jcm, 2024, doi:10.3390/jcm13092487_

Round 1

Reviewer 1 Report

Comments and Suggestions for Authors

The manuscript can be accepted after the authors correct the following comments:

1.     The title is too long and make the reader confuse, so, please revise the title to be concise and clear, ensuring it accurately represents the content.

2.     The abstract section contains duplicate sentences. Please eliminate any unnecessary repetition and ensure that the objective of this study is unambiguous and easily comprehensible.

3.     It is important to observe that scientific publications should contain a minimum of three and a maximum of five keywords.

4.     The introduction part lacks in terms of paragraphs discussing the various sorts of strokes and their classifications. Furthermore, kindly write an academic paragraph that provides details regarding blood arteries, including the circle of Willis and the common carotid artery (CCA).

5.     The caption of figure 1 is too long. Please shorten it.

6.     The manuscript need overall improvement in academic writing.

7.     There are numerous grammatical and typographical problems that require correction.

8.     Some headings are long and this make the readers confuse.

  1. The conclusion section need rewrite to reflects the article information in a good way.

  1. Make sure that all sentences are linked together.

  1. The result of similarity report is 28%. Thus, the authors should reduce it to be less than 16%.

 Please see additional comments in attached file.

Comments on the Quality of English Language

Moderate editing of English language required

Author Response

Answeres for Reviewer 1

Thank you for your detailed review. We have made minor and major changes in the text. We hope it make clearer and more relevant our review. Our answeres are listed after your comments below. 

  1. The title is too long and make the reader confuse, so, please revise the title to be concise and clear, ensuring it accurately represents the content.

We have changed the title: Non-invasive tools in perioperative stroke risk assessment for asymptomatic carotid artery stenosis with a focus on the Circle of Willis 

  1. The abstract section contains duplicate sentences. Please eliminate any unnecessary repetition and ensure that the objective of this study is unambiguous and easily comprehensible.

 The abstract has been modified to be clearer and plainer.

  1. It is important to observe that scientific publications should contain a minimum of three and a maximum of five keywords.

 Keywords have been changed and reduced to 5.

  1. The introduction part lacks in terms of paragraphs discussing the various sorts of strokes and their classifications. Furthermore, kindly write an academic paragraph that provides details regarding blood arteries, including the circle of Willis and the common carotid artery (CCA).

We have added a short general description of stroke. The blood arteries and Cow were described in a latter section (2.1.).

  1. The caption of figure 1 is too long. Please shorten it.

The caption has been shortened.

  1. The manuscript needs overall improvement in academic writing.

 Language editing was performed by a professional.

  1. There are numerous grammatical and typographical problems that require correction.

  Language editing was performed by a professional. American English was used.

  1. Some headings are long and this make the readers confuse

The headings are shortened.

  1. The conclusion section needs rewrite to reflect the article information in a good way.

The conclusion section has been modified to be clearer and plainer. Future direction paragraph has been added before the conclusion section.

  1. Make sure that all sentences are linked together. The result of similarity report is 28%. Thus, the authors should reduce it to be less than 16%.

 Turnitin software was used to check a similarity report by us. The highest rates of the similarities were linked to the previous publications of the authors of this review and these sentences were indicated as references as well in the text. The text has been rephrased in some parts to reduce the similarities.

 Additional comments in attached file.

  1. The abstract part lacks the originality of this investigation.

The abstract has been modified to be clearer and plainer.

  1. In the introduction section, the author referred to ischemic stroke that is linked to vulnerable plaque. Please provide a justification and specify the reasons behind this statement.

We have added a short general description of stroke and added an explanation of plaque vulnerability and ischemic stroke risk. The perioperative neurological event risk is mostly related to impaired cerebrovascular reserve capacity caused hypoperfusion related ischemia or procedure (and not plaque) related embolization.  In our review we focus on the mechanism of peri-operative neurological events caused by impaired cerebrovascular reserve capacity induced new ischemic lesions.    

  1. Table 1 does not need the inclusion of previous investigations.
  2. It is more advantageous to summarize your research findings in a table, as it allows the reader to save time and effort.

Table 1 is a summary of the previous investigations and shows the differences of the CoW segment classification, which has a serious impact on the conflicting results of the different studies. We have added a sentence to explain and emphasize the importance of the nomenclature of the different studies. 

Reviewer 2 Report

Comments and Suggestions for Authors

Dear Authors,

I have read your paper, which aim was to summarize the possible non-invasive pre- and intraoperative additional diagnostic methods, which could be able to predict the potential perioperative ischemic stroke risk based on CoW morphology.

The paper is literature review. It is thorough and covers the aim. The CoW morphology is well presented, as well as imaging modalities used in this entity. There is scientific merit and clinical interest for the readers.

However, I would suggest following improvements:

- Add Methods section and describe how you investigated this topic. Which databases you used, which keywords you typed. Which articles you included and which not and why? Also, articles in which languages were included?

- After adding Methods section with all of aforementioned data, also add Flow chart. In this way you can explain inclusion/exclusion criteria and number of finally included article.

- In the Last paragraph before Conclusion, add Future directions paragraph.

- Correct grammar mistakes.

I suggest revision.

Comments on the Quality of English Language

Spelling and grammar minor editing is needed

Author Response

Answeres for Reviewer 2

Thank you for your review. We have made minor and major changes in the text, but the methods section is not required for a review article, so we had added only explanation and short description for literature search to the supplement section. Our answeres for your comments are below.

  1. Add Methods section and describe how you investigated this topic. Which databases you used, which keywords you typed. Which articles you included and which not and why? Also, articles in which languages were included?
  2. After adding Methods section with all of aforementioned data, also add Flow chart. In this way you can explain inclusion/exclusion criteria and number of finally included article.

This very specific topic is evolving rapidly, but there is not enough publication in this very specific field for even a systematic review or for a proper meta-analysis.  It is a narrative review article, not a systematic review. According to the journal of clinical medicine instruction for authors in case of a review article to add methods section is not compulsory.

  1. In the Last paragraph before Conclusion, add Future directions paragraph.

The conclusion section has been modified to be clearer and plainer. Future direction paragraph has been added to the conclusion section.

  1. Correct grammar mistakes.

  Language editing were performed by a professional. American English was used. If there is a need for further language edition it will be performed by the MDPI official language editor service, which requires a couple of days.

Round 2

Reviewer 1 Report

Comments and Suggestions for Authors

The authors make good corrections.

Comments on the Quality of English Language

Minor editing of English language required

Reviewer 2 Report

Comments and Suggestions for Authors

Dear authors,

thank you for revising the paper according to suggestions.